# Simultaneous Analysis of Flumethasone Pivalate and Clioquinol in the Presence of Phenoxyethanol Preservative in Their Pharmaceuticals Using TLC and UHPLC Methods

Faisal K. Algethami [1] , Huda Salem AlSalem [2] , Mohammed Gamal [3,*] , Nada Nabil [4], Hala E. Zaazaa [5], Mohamed A. Ibrahim [5] and Asmaa A. Mandour [6]

[1] Department of Chemistry, College of Science, Imam Mohammad Ibn Saud Islamic University (IMSIU), Riyadh 11623, Saudi Arabia; falgethami@imamu.edu.sa

[2] Department of Chemistry, College of Science, Princess Nourah bint Abdulrahman University, Riyadh 11671, Saudi Arabia; husalsalem@pnu.edu.sa

[3] Pharmaceutical Analytical Chemistry Department, Faculty of Pharmacy, Beni-Suef University, Alshaheed Shehata Ahmad Hegazy St., Beni-Suef 62514, Egypt

[4] Analytical Chemistry Department, Faculty of Pharmacy, Badr University in Cairo (BUC), Entertainment Area, Cairo 11829, Egypt; nada.nabil@buc.edu.eg

[5] Analytical Chemistry Department, Faculty of Pharmacy, Cairo University, Kasr El-Aini Street, Cairo 11562, Egypt; hala.zaazaa@pharma.cu.edu.eg (H.E.Z.); mohammed.abdelkawy@pharma.cu.edu.eg (M.A.I.)

[6] Pharmaceutical Chemistry Department, Faculty of Pharmacy, Future University in Egypt (FUE), 90th Street, Fifth Settlement, New Cairo 11835, Egypt; asmaa.abdelkereim@fue.edu.eg

[*] Correspondence: mohammed.gamal@pharm.bsu.edu.eg or mgamalm3000@yahoo.com

**Abstract:** Two novel separation methods have been presented for the concurrent assessment of flumethasone pivalate (FP) and clioquinol (CL) in their combinations in ear drop formulations or in the presence of phenoxyethanol preservative (PEP) in their cream formulations. The first method is an innovative thin-layer chromatographic (TLC) method. The optimal separation was accomplished via silica gel aluminum plates F254, with a mixture of benzene, ethyl acetate and formic acid (5:5:0.2, in volumes) as the mobile system. In Method II, a new ultra-high-performance liquid chromatographic method (UHPLC) with a photodiode array detector (PDA) was presented. A reversed-phase inertsil ODS 5 μm C 18 packed column (100 Å, 4.6 mm internal diameter (I.D.) × 50 mm) at 30 °C was employed. Elution was completed in 3 min. Unfortunately, greener solvents were tested as a mobile phase, but an asymmetric peak for CL was noted. In addition, the new UHPLC method has a priority over the old HPLC one by Sayed et al., 2014, in terms of quickness and avoiding interference from the PEP preservative. Concerning the TLC method, the novel TLC method has the advantage of preventing the interference of PEP. This paper represents the first analytical approach for the concurrent assay of FP and CL in the presence of the preservative phenoxyethanol in the cream formulation.

**Keywords:** flumethasone pivalate; clioquinol; TLC; UHPLC; preservative

## 1. Introduction

Flumethasone pivalate (FP), (6S,8S,9R,10S,11S,13S,14S,16R,17R)-6,9-Difluoro-11,17-dihydroxy-17-(2-(2,2-dimethylpropinyl)oxyacetyl)-10,13,16-trimethyl-6,7,8,11,12,14,15,16-octahydrocyclopenta[a]phenanthren-3-one (Figure 1a) [1,2]. FP shows moderate corticosteroid potency. Its topical uses proved its efficacy as an antiallergic, antipruritic, anti-inflammatory and vasoconstrictive agent [3].

Clioquinol (CL), 5-chloro-7-iodo-8-quinolinol (Figure 1b) [1,2]. CL is a member of the hydroxyquinoline family, which inhibits specific enzymes related to the replication of DNA. It is an antiprotozoal and antifungal drug [4]. Both FP and CL are present

together in combination cream (Locacorten Vioform Cream® RIEMSER Pharma GmbH, Greifswald, Germany) and ear drop (Viotic Ear Drops® AMOUN Pharmaceuticals, Cairo, Egypt)formulations.

**Figure 1.** Chemical structure of (**A**) flumethasone pivalate and (**B**) clioquinol.

Phenoxyethanol, or 2-phenoxyethanol, is the most commonly used preservative in cream dosage and is susceptible to microbial growth. The main role of the preservative is to avoid any degradation or alteration of the product and, hence, prolong its shelf life [5]. Phenoxyethanol preservative (CAS No. 122-99-6) is reported in Annex V/29 of the Cosmetics Regulations for the European Commission, Scientific Committee on Consumer Safety, Phenoxyethanol No. 1223/2009 [6]. Phenoxyethanol is commonly added as a preservative in cream pharmaceuticals at a concentration of 1% [6]. If it is used in concentrations over 1%, phenoxyethanol is harmful in skin cosmetics [6].

According to the latest literature investigations, many analytical approaches have been established for the assay of both drugs, either individually or in combination with other drugs. FL and CL have been previously studied individually by HPLC [7–12]. FP has been previously investigated by HPLC in combination with salicylic acid [13]. Additionally, CL has been previously analyzed with betamethasone valerate [14] and hydrocortisone [15] by HPLC. However, the literature survey reveals that only one study was recorded for the synchronized assay of FP and CL by HPLC and TLC methods in their ear drop formulation [16]. Additionally, FP and CL were concurrently determined using variable UV spectrophotometric techniques, including ratio subtraction, ratio difference spectrophotometric methods, dual-wavelength, area under the curve and first derivative ratio spectrophotometric methods [17,18].

Preservatives are essential additives in pharmaceutical formulations to retain them at acceptable quality and safety standards during shelf life. Preservatives inhibit micro-organisms growth and their potential toxicities. However, their presence should not reduce the safety, efficacy or bioavailability of the final pharmaceutical. Additionally, they should not interfere with the main active medicines during their analysis [19]. Thus, this work aimed to analyze FP and CL in the presence of a low-concentration preservative, namely 2-phenoxyethanol, in cream formulations using two novel and valid chromatographic approaches. It is well known that TLC and HPLC approaches can save time and money in quality control (QC) laboratories during the daily analysis of medicines. Additionally, UHPLC is designed with small particle stationary phases and short columns for achieving the recommended fastness during regular analysis in QC units. UHPLC offers perfect performance if compared to the traditional HPLC instrument [20].

Up to date, no analytical approaches have been stated for the concurrent assay of flumethasone pivalate and clioquinol in the presence of the preservative phenoxyethanol in

the cream dosage form. Thus, the principal goal of this analytical paper is to establish novel, economical, simple, selective, precise and validated analytical methods for simultaneous determination of flumethasone pivalate and clioquinol in the presence of phenoxyethanol preservative in the cream dosage form and also in their dual mixture in ear drops. Validation items for the novel chromatographic methods were monitored according to the International Conference for Harmonization (ICH) strategies [21]. Additionally, the goals were extended to check the efficacy, safety and quality of the aforementioned pharmaceuticals.

## 2. Materials and Methods

### 2.1. Description for Instruments

Aluminum TLC-plates silica gel-coated F254 (20 × 20 cm), 0.20 mm thickness, Fluka (Buchs, Switzerland) were utilized. Camag Linomat 5 autosampling (Muttens, Switzerland) fortified with a (100 μL) microsyringe was utilized for applying specimens at a persistent speed of 10 μL in a second as 6 mm bands. TLC-densitometric Scanner Camag model 3S/N 130,319 in the reflecting absorption manner (Muttens, Switzerland) with a 20 mm per second speed of scanning and connected to Win CATS software (Muttens, Switzerland). The slit size was retained at 6.00 × 0.30 mm. Visualizing drug spots was performed via a UV lamp of 254 nm wavelength (Georgia, USA).

A 1290 infinity Agilent ultra-high performance liquid chromatography connected to a 1290 Diode array detector(California, USA), Automated Liquid Sampler (ALS), Thermostated Column Copartment (TCC) for the column, controllable quaternary pump Vertical In-Line Close (VL) and 1290 Thermostat. An Agilent Chemstation (B.04.03) and Lab Advisor (Utility) Quantitative analysis (B.02.04) programs were used for data procurement and processing. Separation was conducted on reversed phase inertsil ODS 5 μm C 18 stationary phase (4.6 × 50 mm, 100 Å) at room temperature, and isocratic elution was attained by acidic buffer pH 3 of phosphate type (having 100 mg Heptane-1-sulphonic acid sodium salt per 100 mL) and acetonitrile (35:65, by volumes). The injected volume was one microliter. The detection mode was a photodiode array detector (PDA).

### 2.2. Reagents and Chemicals

Flumethasone pivalate of 99.7% purity and clioquinol of 99.92% purity standards were purchased from AMOUN Pharmaceuticals (Cairo, Egypt). Phenoxyethanol 94% (preservative) was purchased from Thermo Fisher (Bremen, Germany).

Methyl alcohol, acetonitrile, Heptane-1-sulfonic acid sodium salt, ortho-phosphoric acid, Na H2 phosphate, ethyl acetate, benzene and formic acid were purchased from Sigma Aldrich, Gillingham, UK. All chemicals used were of HPLC purity. An Elga Ultrapure Q apparatus, Oxford, UK was applied for ultra-pure water purification. Phosphate buffer was prepared via dissolution of 3.39 g of sodium phosphate monobasic in one liter of ultra-pure water and the final pH was monitored at 3 using ortho-phosphoric acid.

Stock solutions of FP and CL were prepared in methyl alcohol at 1.00 mg/mL for TLC and UHPLC. The stock solution of phenoxyethanol was prepared in methyl alcohol at 0.50 mg/mL for TLC and UHPLC.

Concerning Method I (MI); the TLC-densitometric approach. Amounts corresponding to 2.00–12.00 mg of FP and 2.00–10.00 mg of CL were moved from their corresponding parent flasks into two distinct series of 10-mL glass flasks, and then the flasks were filled with methyl alcohol.

Concerning Method II (MII); the UHPLC method. In many 10-mL glass flasks, aliquots corresponding to 0.05–0.50 mg FP and 0.05–0.60 mg CL were, separately, moved from their corresponding standard liquids and quantitatively diluted with the liquid phase.

### 2.3. Pharmaceutical Formulations

Locacorten Vioform Cream; each gram labeled to have 0.2 mg/g FP and 30 mg/g CL (Batch No: 701980); produced by RIEMSER Pharma GmbH, was purchased from the market in Germany.

Viotic ear drop[®]; labeled to have 0.2 mg FP and 10 mg CL per mL (Batch no: 192059); manufactured by AMOUN Pharmaceutical, was purchased from the community pharmacies in Egypt.

### 2.4. Analytical Procedures

Chromatographic Environments and Establishment of Linearity

Method I (MI); the TLC-densitometric approach. Ten microliters from each working solution were applied to 20 × 10 cm TLC plates in triplicate (the size of the band was 6 mm; spacing between each two successive bands was 14 mm; the distances from the sides and the bottom edge of the plate were 10 and 15 mm) via a Camag Linomat auto sampling. The separation tank was saturated for 20 min with the developing system containing benzene: ethyl acetate: formic acid (5:5:0.2, by volume) at 25 °C, while the actual analysis time was 3 min. The developed TLC plates were dried with the aid of fresh air and scanned at 250 nm. Lastly, the linearity and regression equations were developed by graphing the average peak areas against the equivalent concentrations.

Method II (MII); the UHPLC method. One microliter from each solution was injected three times and chromatographed on reversed phase C-18 column with the aforementioned data, and the elution was completed by buffer pH 3 of phosphate type (having 0.1 g heptane-1-sulfonic acid sodium salt per 100 mL) and acetonitrile (35:65, by volume). The peak areas were recorded, and regression equations and the calibration curves were developed.

### 2.5. Application to Pharmaceutical Formulations

#### 2.5.1. Viotic[®] Ear Drops

The recorded procedures in the old HPLC method [16] were followed for FP and CL analysis in their ear drops formulation.

Concerning MI, five milliliters of Viotic[®] ear drops (one mL labeled to have 0.20 mg FP and 10.00 mg CL) were taken into a 10-mL glass flask, then methyl alcohol was added to complete the final volume and sonicated for 10 min (0.10 mg per mL of FP and 5.00 mg per mL of CL). The solutions were filtered via a 0.22 µm syringe before usage. Aliquots of 20.00 µL of this solution were spotted to determine FP, while CL was determined by diluting 0.4 milliliters of this solution with methyl alcohol into a 10-mL glass flask; followed by spotting of 20.00 µL of this diluted solution. TLC plates were developed and scanned. Then, the concentrations of FP and CL were calculated from their equivalent scanned peak areas.

Regarding MII, five milliliters of Viotic[®] ear drops with the above-mentioned concentrations of FP and CL were taken into a 25-mL flask, then methyl alcohol was poured to the final size and sonicated for 10 min (0.04 mg/mL of FP and 2.00 mg/mL of CL). The solutions were filtered via a 0.22 µm before usage. Three milliliters of this liquid were diluted into a 10-mL glass flask with methyl alcohol for FP determination, while CL was determined by diluting 0.10 mL of this liquid solution with methyl alcohol into a 10-mL glass flask. One microliter of each solution was injected three times.

#### 2.5.2. Locacorten Vioform Cream[®]

The procedures for drug extraction from cream dosage [22] were applied as follows:

An accurate weight (2.5 g) of Locacorten Vioform Cream[®] (each one gram branded to have 0.2 mg of FP and 30 mg of CL) was placed into a 50-mL beaker, 4 mL of methyl alcohol was added and the temperature was controlled at 60 °C using a water bath with continuous stirring till the cream was completely melted, then the solution was cooled to solidify the base, the methyl alcohol layer was decanted into 10-mL glass flask, three extraction processes were conducted. Then, the united extracts were either diluted with methyl alcohol to 10-mL final volume (0.05 mg/mL of FP and 7.50 mg/mL CL) (MI). Aliquots of 40.00 µL of this solution were spotted to determine the FP, while CL was determined by diluting 0.4 µL of this solution with methyl alcohol into a 10-mL volumetric flask; followed by spotting 10.00 µL.

Concerning (MII), 5 g of the cream were added to a 50-mL volumetric flask with methyl alcohol (0.02 mg/mL of FP and 3.00 mg/mL of CL). Then, five milliliters of this liquid were diluted into a 10-mL glass flask with methyl alcohol for FP determination, while CL was determined by diluting 0.25 mL of this liquid with methyl alcohol into a 50-mL glass flask.

## 3. Results and Discussions

### 3.1. Method Optimization

#### 3.1.1. Method I (MI)

Different mobile phases of different compositions and proportions were tried to isolate the medicines mentioned from phenoxyethanol. Different systems were tried, such as chloroform:methyl alcohol, ethyl acetate:methyl alcohol:ammonia or glacial acetic acid and hexane or benzene: acetone in altered percentages, but no optimal resolution was achieved.

The best-developed system was ethyl acetate:benzene:formic acid (5:5:0.2 by volume) in terms of optimal peak symmetry and highest achieved resolution for the aforementioned drugs. The detailed percentages for the tested mobile systems were listed in Supplementary Table S1. This selected developing system allows the finest resolution of the three compounds and their quantitative estimation without any noted interference (Figure 2). Furthermore, 6 mm was the finest band dimension that allowed for well-defined, sharp peaks. Additionally, 250 nm was the ideal wavelength in terms of sensitivity and noise reduction.

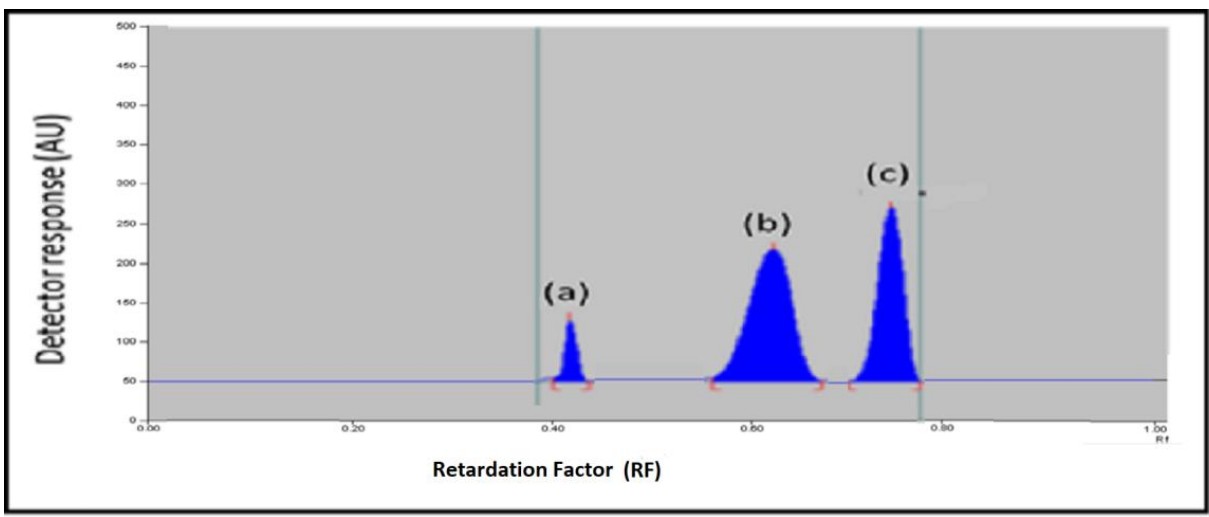

**Figure 2.** TLC densitogram of (a) 4 μg/band of clioquinol (Rf 0.41), (b) 5 μg/band phenoxyethanol (Rf 0.61) and (c) 8 μg/band flumethasone pivalate (Rf 0.74); the developing system consists of benzene:ethyl acetate:formic acid (5:5:0.2, by volume), at 250 nm.

#### 3.1.2. Method II (MII)

The reversed phase inertsil ODS 5μm C 18 stationary phase with the aforementioned dimensions was selected for separation and quantitation of FP and CL based on the outcomes of Sayed et al., 2014 [16]. Additionally, greener solvents, e.g., water with ethanol, were tested as a liquid mobile phase in diverse volumes, but very poor resolution and an asymmetric peak for CL were attained. Phosphate buffers (as the aqueous system) in different pH ranges were tested with methyl alcohol or acetonitrile at altered percentages, but poor resolution and asymmetric peaks were achieved. Controlling the pH at three via phosphate buffer having 0.1 g heptane-1-sulfonic acid sodium salt per 100 mL: acetonitrile (35:65, by volumes), satisfactory separation for both medicines of FP and CL and the ternary mixture of FP, CL in the coexistence of phenoxyethanol was achieved with accepted resolution and a short chromatographic time, where the retention time (Rt) values of phenoxyethanol, FP and CL were 0.7, 1.8 and 2.8 min (Figure 3a,b).

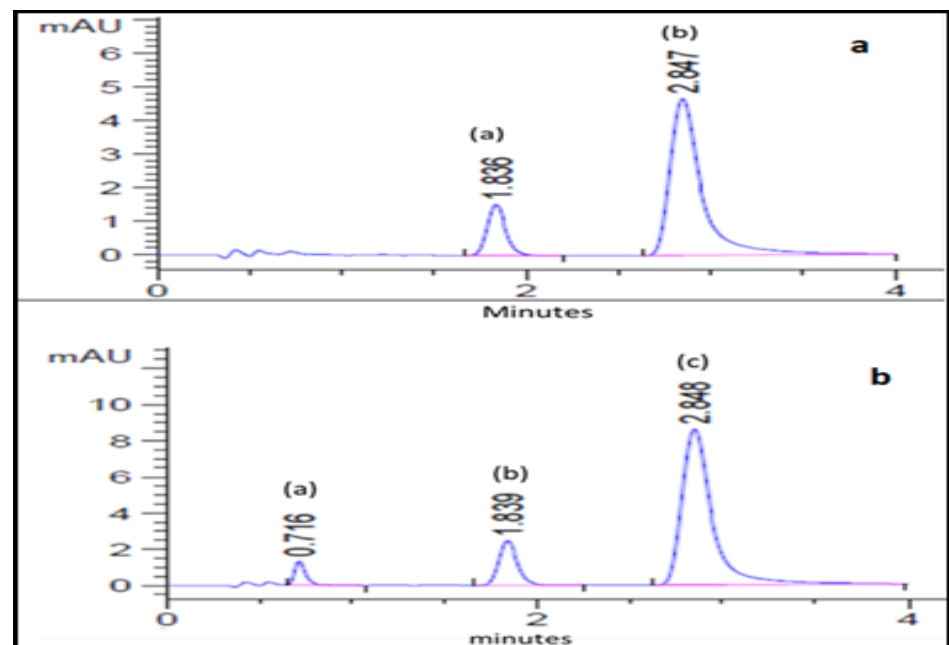

**Figure 3.** (**a**) UHPLC chromatogram of a lab-prepared mixture containing (a) 10 μg/mL flumethasone pivalate retention time (Rt) = 1.8 and (b) 30 μg/mL clioquinol (Rt = 2.8) on an inertsil ODS shield C18 column, mobile phase consists of phosphate buffer pH 3 acetonitrile (35:65, by volume), red line represents baseline while blue one represents eluted peaks. (**b**) UHPLC chromatogram of a Lab prepared mixture containing (a) 5 μg/mL phenoxy ethanol (Rt = 0.7), (b) 10 μg/mL flumethasone pivalate (Rt = 1.8), and (c) 30 μg/mL clioquinol (Rt = 2.8) on Inertsil ODS shield C18 column, mobile phase consists of phosphate buffer pH 3 acetonitrile (35:65, by volume).

Heptane-1-sulfonic acid Na salt is used as an ion pair coupling chemical added to the mobile phase to enhance the separation of ionic analytes by promoting the formation of ion pairs via ionic interaction, rendering the sample more hydrophobic in reversed-phase chromatography and hence analytes are eluted more gradually. The hydrophobic region of the ion pair reagent also allows the interaction with the stationary phase to achieve the optimal peak outline of CL, which results in convincing resolution with the least separation time [23].

### 3.2. Methods Validation

ICH protocols were applied during the validation procedures [19]. The outcomes illustrated in (Table 1) were very convincing. The noted linear ranges were 2.00–12.00 and 2.00–10.00 μg/band for FP and CL, correspondingly for the TLC densitometric method (Table 1), while the recorded ranges were 5.00–50.00 and 5.00–60.00 μg/mL for FP and CL, correspondingly for the UHPLC method (Table 1).

**Table 1.** Validation parameters of the novel TLC-densitometric and RP-UHPLC approaches for the assay of flumethasone pivalate and clioquinol.

| Items | TLC Method | | RP-UHPLC Method | |
|---|---|---|---|---|
| | **Flumethasone Pivalate** | **Clioquinol** | **Flumethasone Pivalate** | **Clioquinol** |
| Wavelength (nm) | 250 | | 250 | |
| Analysis speed (minutes) | 10 | | 3 | |
| | Regression parameters | | | |
| Working range | 2.00–12.00 μg/band | 2.00–10.00 μg/band | 5.00–50.00 μg/mL | 5.00–60.00 μg/mL |
| Intercept | +5604.9 | +365.92 | −0.3079 | −7.9177 |

**Table 1.** *Cont.*

| Items | TLC Method | | RP-UHPLC Method | |
|---|---|---|---|---|
| | **Flumethasone Pivalate** | **Clioquinol** | **Flumethasone Pivalate** | **Clioquinol** |
| Slope | 1407.6 | 246.92 | 0.8637 | 5.1626 |
| Correlation coefficients | 0.9998 | 0.9997 | 0.9999 | 0.9999 |
| Accuracy of mean ± standard deviation (SD) | 100.01 ± 0.91 | 100.09 ± 0.99 | 99.62 ± 0.77 | 99.93 ± 1.21 |
| Precision (±% relative Standard deviation (RSD) Intraday precision [a] | ±0.42 | ±0.72 | ±0.54 | ±0.71 |
| Precision (±%RSD) Intermediate precision [b] | ±1.19 | ±1.43 | ±1.05 | ±0.83 |
| Specificity [c] (mean ± SD) | 100.02 ± 0.57 | 100.17 ± 0.40 | 100.42 ± 0.47 | 99.39 ± 0.38 |
| Robustness | 99.97 ± 0.96 | 100.09 ± 1.04 | 100.06 ± 0.73 | 100.34 ± 0.39 |
| Limit of detection (LOD) [d] | 0.52 μg/band | 0.63 μg/band | 1.52 μg/mL | 1.47 μg/mL |
| Limit of quantitation (LOQ) [d] | 1.57 μg/band | 1.91·μg/band | 4.63 μg/mL | 4.45 μg/mL |

[a] In the day precision: the percentage of relative SD for 3 numerous concentrations (2.5, 4.5 and 5.5 μg/band for FP and 2.5, 3.5 and 4.5 μg/band for CL) for TLC-densitometric and (12.5, 22.5 and 32.5 μg/mL for FP and CL) for RP-UHPLC/3 repeats each, in the exact day. [b] Interday precision: the percentage of relative SD of 3 dissimilar concentrations (2.5, 4.5 and 5.5 μg/band for FP and 2.5, 3.5 and 4.5 μg/band for CL) for TLC-densitometric and (12.5, 22.5 and 32.5 μg/mL for FP and CL) for RP-UHPLC/3 repeats each on 3 uninterrupted days. [c] Recoveries of FP and CL in laboratory mixtures in the presence of the preservative phenoxyethanol. The mean percent recovery (%R) for triplicate determinations of the aforementioned three concentration levels of each drug was calculated. [d] Calculated from the equations [LOD =3.3 (standard deviation/slope), LOQ = 3 × LOD].

Specificity was assured via the accepted resolutions between the three components FP, CL and the preservative phenoxyethanol, as demonstrated in Figures 2–4.

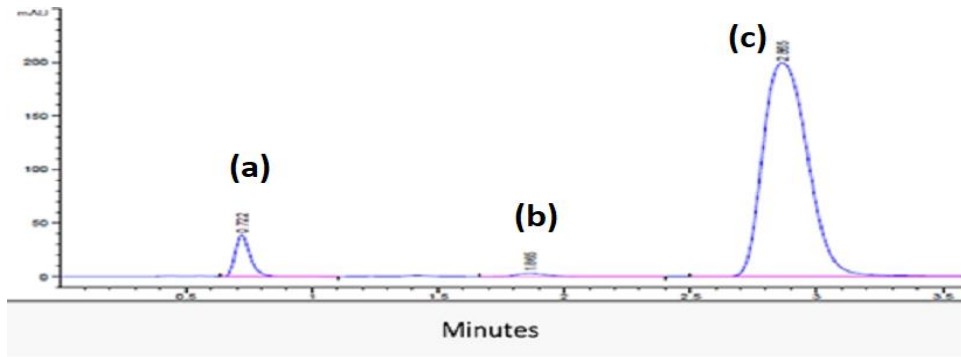

**Figure 4.** UHPLC chromatogram of Locacorten Vioform Cream® (a) phenoxyethanol (Rt = 0.7), (b) flumethasone pivalate (Rt = 1.8) and (c) clioquinol (Rt = 2.8), on an inertsil ODS shield C18 column, mobile phase consists of phosphate buffer pH 3 and acetonitrile (35:65, by volume).

The accuracy of the innovative methods was evaluated by applying the standard addition technique to ear drops and cream where known concentrations of FP and CL have been added at different concentration levels (Tables 2 and 3). The standard addition technique is performed by adding the amount of the standard drug to the dosage form. The actual and added concentration values are stated in Tables 2 and 3 as well.

**Table 2.** Assessment of flumethasone pivalate and clioquinol in Viotic® ear drops by the innovative chromatographic methods and outcomes of standard addition techniques.

| Pharmaceutical | Drugs | TLC–Densitometric Method (Standard Addition) | | | | | RP-UHPLC Method (Standard Addition) | | | | |
|---|---|---|---|---|---|---|---|---|---|---|---|
| | | Claimed Taken | Added | Total Found [b] | Standard Found [b] | %Recoveries of Added [b] | Claimed Taken | Added | Total Found [b] | Standard Found [b] | %Recoveries of Added [b] |
| Units | | In (µg/band) | | | | | In (µg/mL) | | | | |
| Viotic® ear Drops [a] B.N (192059) | FP | 2.00 | - | 1.97 | - | - | 12.00 | - | 12.18 | - | - |
| | | 2.00 | 3.00 | 4.93 | 2.96 | 98.66 | 12.00 | 6.00 | 18.11 | 5.93 | 98.83 |
| | | 2.00 | 4.00 | 6.01 | 4.04 | 101.00 | 12.00 | 12.00 | 24.02 | 11.84 | 98.66 |
| | | 2.00 | 5.00 | 7.03 | 5.06 | 101.20 | 12.00 | 15.00 | 27.22 | 15.04 | 100.20 |
| | | Mean ± SD [b] | | | | 100.28 ± 1.41 | Mean ± SD [b] | | | | 99.23 ± 0.84 |
| | CL | 4.00 | - | 4.04 | - | - | 20.00 | - | 19.91 | - | - |
| | | 4.00 | 2.00 | 6.00 | 1.96 | 98.00 | 20.00 | 10.00 | 30.05 | 10.14 | 101.40 |
| | | 4.00 | 4.00 | 8.07 | 4.03 | 100.75 | 20.00 | 20.00 | 40.30 | 20.39 | 101.95 |
| | | 4.00 | 5.00 | 8.96 | 4.92 | 98.40 | 20.00 | 25.00 | 44.93 | 25.02 | 100.08 |
| | | Mean ± SD [b] | | | | 99.05 ± 1.48 | Mean ± SD [b] | | | | 101.14 ± 0.96 |

[a] Labeled to have 0.2 mg FP and 10 mg CL. [b] Average for three measurements.

**Table 3.** Assessment of flumethasone pivalate and clioquinol in Locacorten Vioform Cream® by the new TLC and RP-UHPLC methods and results for the standard addition techniques.

| Product | Drugs | TLC–Densitometric Method (Standard Addition) | | | | | RP-UHPLC Method (Standard Addition) | | | | |
|---|---|---|---|---|---|---|---|---|---|---|---|
| | | Claimed Taken | Added | Total Found [b] | Standard Found [b] | Recoveries Percentage for Added [b] | Claimed Take | Added | Total Found [b] | Standard Found [b] | Recoveries Percentage for Added [b] |
| Locacorten Vioform Cream® [a] B.N (701,980) | FP | 2.00 | - | 2.01 | - | - | 10.00 | - | 9.96 | - | - |
| | | 2.00 | 3.00 | 4.95 | 2.94 | 98.00 | 10.00 | 5.00 | 14.94 | 4.98 | 99.60 |
| | | 2.00 | 4.00 | 5.96 | 3.95 | 98.75 | 10.00 | 10.00 | 19.92 | 9.96 | 99.60 |
| | | 2.00 | 5.00 | 7.05 | 5.04 | 100.80 | 10.00 | 20.00 | 29.78 | 19.82 | 99.10 |
| | | Mean ± SD [b] | | | | 99.18 ± 1.44 | Mean ± SD [b] | | | | 99.43 ± 0.28 |
| | CL | 3.00 | - | 3.01 | - | - | 15.00 | - | 15.09 | - | - |
| | | 3.00 | 2.00 | 5.02 | 2.01 | 100.50 | 15.00 | 7.00 | 22.06 | 6.97 | 99.57 |
| | | 3.00 | 3.00 | 5.96 | 2.95 | 98.33 | 15.00 | 15.00 | 30.12 | 15.03 | 100.20 |
| | | 3.00 | 4.00 | 6.98 | 3.97 | 99.25 | 15.00 | 30.00 | 45.38 | 30.29 | 100.96 |
| | | Mean ± SD [b] | | | | 99.36 ± 1.08 | Mean ±SD [b] | | | | 100.24 ± 0.69 |

[a] Labeled to have 0.2 mg FP and 30 mg CL. [b] Average of three measurements. Concentrations for the TLC method were in (µg/band) while for UHPLC methods were in (µg/mL).

Furthermore, the robustness of the two methods was considered by investigating the influence of slight changes in the experimental environments on the method suitability factors. Concerning the TLC approach, robustness was tested under altered environments, such as the development length in cm ($17.00 \pm 1.00$ cm), the developing mobile liquid volume ($80.00 \pm 10.00$ mL) and the saturation time of the analysis tank ($20.00 \pm 2.00$ min), where the calculated $R_f$ ratios for the medicines were the same and the resultant resolutions (Rs) were at all times convenient, assuring the reliability of the TLC method (Table 4). System suitability items were investigated, e.g., capacity, selectivity and tailing factor, where good outcomes were acquired [24] (Table 5). For the RP-UHPLC approach, the three analytes were well resolved under a variety of settings via many flow speeds ($1 \pm 0.2$ mL/min), altered pHs ($3 \pm 0.1$) and diverse temperatures ($30$ °C $\pm 2$). The recorded values for retention times (Rts) of the medicines illustrated in Figure 3a were relatively the same in all cases (Table 6), except for the flow speed, where slight changes in Rts were recorded. However, the calculated resolutions (Rs) were each time not less than 1.5, illustrating accepted chromatograms. Outcomes for capacity ($K'$) and tailing (T) factors were in alignment with international standards [25] (Table 6). Data for system suitability items for the UHPLC method is displayed in Table 7.

**Table 4.** Detailed study for ensuring the robustness of the innovative TLC method.

| Drug | Parameters | | T [a] | $K'$ [b] | Rs [c] | Assay Percentage [d] |
|---|---|---|---|---|---|---|
| CL | Developing liquid phase amount | 80 + 10 mL | 0.87 | 1.27 | - | 98.57 |
| | | 80 − 10 mL | 1.00 | 1.27 | - | 99.71 |
| | Duration of saturation of chromatographic tank | 20 + 2 min | 0.80 | 1.27 | - | 100.28 |
| | | 20 − 2 min | 0.83 | 1.30 | - | 101.71 |
| | Development distance | 17 + 1 cm | 0.87 | 1.27 | - | 99.71 |
| | | 17 − 1 cm | 0.91 | 1.26 | - | 100.57 |
| FP | Developing system amount | 80 + 10 mL | 0.83 | 0.33 | 1.43 | 99.11 |
| | | 80 − 10 mL | 1.00 | 0.35 | 1.42 | 99.42 |
| | Duration of the saturation of developing tank | 20 + 2 min | 1.00 | 0.36 | 1.43 | 101.10 |
| | | 20 − 2 min | 1.11 | 0.37 | 1.44 | 100.56 |
| | Development distance | 17 + 1 cm | 1.05 | 0.36 | 1.46 | 98.84 |
| | | 17 − 1 cm | 1.00 | 0.33 | 1.45 | 100.83 |

[a]. Tailing factor (T) was calculated as the ratio of back to front width at 10% of peak height. [b], $K' = (1 - R_f)/R_f$. [c], Rs = $R_{f2} - R_{f1}/0.5$ (W1 + W2). [d] = average of calculated concentration for 3 estimations X 100/actual concentration.

**Table 5.** Considerations for system suitability evaluation for the innovative TLC method.

| Parameters | CL | Phenoxy Ethanol | FP | Reference Value [24] |
|---|---|---|---|---|
| $K'$ "capacity factor" | 1.32 | 0.63 | 0.35 | The higher $K'$, the smaller the retardation factor |
| $\alpha$ "Relative retention" | | 2.09 | 1.80 | >1 |
| Resolution (Rs) | | 2.40 | 1.40 | >1 |
| Symmetry factor | 1.00 | 1.00 | 0.83 | 1 for ideal peak |

$K' = (1 - R_f)/R_f$. $\alpha = K_2/K_1$. Rs = $R_{f2} - R_{f1}/0.5$ (W1 + W2). Symmetry factor (T) was calculated as the ratio of back to front width at 10% of peak height.

**Table 6.** Detailed study for ensuring the robustness for the new RP-UHPLC method.

| Medicine | Robustness Items | | T [a] | K' [b] | Rs [c] for Peaks of FP and CL | Assay Percentage [d] |
|---|---|---|---|---|---|---|
| FP | Flow speed | 1 + 0.2 mL/min | 0.77 | 17.22 | 3.66 | 99.35 |
| | | 1 − 0.2 mL/min | 0.91 | 16.76 | 3.82 | 99.38 |
| | pH | 3 + 0.1 units | 0.78 | 17.10 | 3.53 | 100.92 |
| | | 3 − 0.1 units | 0.75 | 16.95 | 3.61 | 101.01 |
| | Temp | 30 − 2 °C | 0.80 | 17.20 | 3.41 | 99.83 |
| | | 30 + 2 °C | 0.79 | 16.71 | 3.84 | 99.87 |
| CL | Flow rate | 1 + 0.2 mL/min | 1.31 | 26.43 | 3.78 | 100.45 |
| | | 1 − 0.2 mL/min | 1.29 | 27.09 | 3.66 | 100.84 |
| | pH values | 3 + 0.1 units | 1.31 | 26.53 | 3.75 | 99.79 |
| | | 3 − 0.1 units | 1.35 | 27.05 | 3.76 | 99.95 |
| | Temp | 30 − 2 °C | 1.30 | 27.06 | 3.65 | 100.56 |
| | | 30 + 2 °C | 1.30 | 26.64 | 3.68 | 100.48 |

[a]. Tailing factor (T) was calculated as the ratio of back to front width at 10% of peak height. [b]. $K' = (Rt - Rt_{(0)})/Rt_{(0)}$. [c]. $Rs = R_{t2} - R_{t1}/0.5(W1 + W2)$. [d]. = average of calculated concentration for 3 estimations X 100/actual concentration.

**Table 7.** Considerations for system suitability evaluation for the new RP-UHPLC method.

| Parameters | Obtained Values | | | Reference Values [25] |
|---|---|---|---|---|
| | Phenoxy Ethanol | FP | CL | |
| Resolution (Rs) | 7.12 | 3.71 | | R higher than 2 |
| α "relative retention" | 2.86 | 1.58 | | >1 |
| K' "capacity factor" | 5.93 | 16.96 | 26.76 | K' > 2 |
| N "column efficiency" | 719 | 1205 | 1187 | The higher N, the highly efficient the method |
| symmetry factor | 1.10 | 1.00 | 1.30 | =1 for ideal peak |

$Rs = R_{t2} - R_{t1}/0.5(W1 + W2)$. $α = K2/K1$. $K' = (Rt - Rt_{(0)})/Rt_{(0)}$. N = column length (L)/height equivalent theoretical plates (HETP). Symmetry factor was calculated as the ratio of back to front width at a 10 % of peak height.

The innovative methods were effectively applied for the assessment of FP and CL in cream and ear drop pharmaceuticals. Good recoveries in Tables 2 and 3 were demonstrated for the labeled concentrations and the standard addition protocol was efficiently assured.

Furthermore, the new UHPLC method shows comparable sensitivity when compared with the earlier stated HPLC method by Sayed et al. [16], as stated in detail in Table 8. The new UHPLC method has priority over the earlier stated HPLC method [16] in terms of rapidness, where the analysis time was less than 4 min in the new UHPLC method. Additionally, a smaller volume of acetonitrile was used in the novel UHPLC method. Upon comparison of LOD and LOQ for our novel methods with old, recorded methods in the literature, Table S2, our novel methods showed relatively comparable values with the methods stated by Sayed et al. [16]. However, the maximal sensitivity was observed in the HPLC with electrochemical detection [12] by Bondiolotti et al., 2006, where it detected and quantified CL at the nanogram level in plasma.

The reliability and efficiency of the UV detector coupled with the LC technique are well recognized for the analysis of drug mixtures [26]. Additionally, the TLC method has the merits of robustness, sustainability and multiple assays of many samples concurrently in a short time [27,28].

**Table 8.** Comparisons of the novel RP-UHPLC and TLC methods with the old HPLC and TLC methods.

| Parameters | New UHPLC Method | | | Old HPLC Method by Sayed et al., 2014 [16] | | |
|---|---|---|---|---|---|---|
| Drug Name | Phenoxy Ethanol | Flumethasone Pivalate | Clioquinol | Flumethasone Related Substance | Flumethasone | Clioquinol |
| Range (µg/mL) | not determined | 5.00–50.00 µg/mL | 5.00–60.00 µg/mL | 2.00–35.00 µg/mL | 5.00–50.00 µg/mL | 10.00–70.00 µg/mL |
| Retention time (min) | 0.7 | 1.8 | 2.8 | 2.97 | 6.81 | 10 |
| Mobile phase | Acidic buffer pH 3 of phosphate type (having 0.1 g heptane-1-sulphonic acid sodium salt per 100 mL) and acetonitrile (35:65, by volume). | | | acetonitrile–$H_2O$ (70:30, by volume). | | |
| Stationary phases | Inertsil ODS 5µm C-18 stationary phase (100 Å, 4.6 × 50 mm). | | | a C-18 -ODS (Shimadzu, Japan), 25 cm × 4.6 mm I.D. | | |
| Detection wavelength | 250 nm | | | 235 nm | | |
| Parameters | New TLC method | | | Old TLC method by Sayed et al., 2014 [16] | | |
| Drug name | phenoxy ethanol | flumethasone pivalate | clioquinol | flumethasone related substance | flumethasone | clioquinol |
| Range (µg/mL) | Not determined | 2–12 µg band$^{-1}$ | 2–10 µg band$^{-1}$ | 0.3–4 µg band$^{-1}$ | 0.3–3 µg band$^{-1}$ | 1.5–5 µg band$^{-1}$ |
| Retardation factor | 0.61 | 0.74 | 0.41 | 0.31 | 0.07 | 0.64 |
| Mobile system | A mixture of benzene:ethyl acetate:formic acid (5:5:0.2, by volume) | | | A mixture of benzene:hexane:acetone:formic acid (5:4:2:0.13, by volume) | | |
| Stationary phases | silica gel aluminum plates F254 | | | silica gel aluminum plates 60 F254 | | |
| Detection wavelength | 250 nm | | | 235 nm | | |

## 4. Conclusions

From the above experimental results, the newly validated chromatographic methods provide accurate, precise, reproducible and sensitive methods for the assay and quantification of flumethasone pivalate and clioquinol in the coexistence of the preservative phenoxyethanol for the first time. The innovative TLC and RP-UHPLC approaches were efficiently used for the assessment of flumethasone pivalate and clioquinol in pure powders and their combined pharmaceutical creams and ear drops. Furthermore, the new UHPLC method shows comparable sensitivity when compared with the earlier stated HPLC method by Sayed et al., 2014. The new UHPLC method has priority over the earlier stated HPLC method by Sayed et al., 2014, in terms of rapidness, where the analysis time was less than 4 min in the new UHPLC method. Additionally, a smaller volume of acetonitrile was used in the novel UHPLC method. The newly developed methods were suitable for routine quality control analysis.

**Supplementary Materials:** The following supporting information can be downloaded at: https://www.mdpi.com/article/10.3390/pr11071888/s1, Table S1: Different mobile phases used for optimization of the novel TLC-densitometric the assay of flumethasone pivalate and clioquinol; Table S2: Comparasions for LOD and LOQ for the novel TLC-densitometric and UHPLC methods with refernces methods for the assay of flumethasone pivalate and clioquinol.

**Author Contributions:** F.K.A., H.S.A., M.G., N.N., H.E.Z., M.A.I. and A.A.M.,.: Data curation, Conceptualization, Formal analysis, Investigation, Validation, Writing—original draft and Methodology. N.N., H.E.Z., M.A.I. and A.A.M.: Methodology, Investigation, Resources and Writing—review and editing. All authors have read and agreed to the published version of the manuscript.

**Funding:** Princess Nourah bint Abdulrahman University Researchers Supporting Project number (PNURSP2023R185), Princess Nourah bint Abdulrahman University, Riyadh, Saudi Arabia.

**Data Availability Statement:** The corresponding authors will deliver the required data whenever desired.

**Acknowledgments:** Princess Nourah bint Abdulrahman University Researchers Supporting Project number (PNURSP2023R185),Princess Nourah bint Abdulrahman University, Riyadh, Saudi Arabia.

**Conflicts of Interest:** The authors declare no conflict of interest.

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
