# Peer review of "Simultaneous Analysis of Flumethasone Pivalate and Clioquinol in the Presence of Phenoxyethanol Preservative in Their Pharmaceuticals Using TLC and UHPLC Methods"

_processes, doi:10.3390/pr11071888_

Round 1

Reviewer 1 Report

The monitoring of concentration of special compounds in different formula is important. The manuscript described two directions of analytical processes for flumethasone pivalate and clioquinol. Both methods can be applied satisfactorily in real cream formulation samples. Since TCL and HPLC methods have been reported in many references. It would be interested by some chemists. It can be shorten in the TLC session and more introduction of UHPLC.

no comments here.

Author Response

Replies to reviewers' comments

Great appreciation for all the reviewers and the editor for their productive recommendations. Really, the quality of this paper was greatly improved following their suggestions.   

Reviewer 1

The monitoring of concentration of special compounds in different formula is important. The manuscript described two directions of analytical processes for flumethasone pivalate and clioquinol. Both methods can be applied satisfactorily in real cream formulation samples. Since TCL and HPLC methods have been reported in many references. It would be interested by some chemists. It can be shorten in the TLC session and more introduction of UHPLC.

Thanks for your positive point of view. The following paragraph and new reference 20 was added.

Besides, UHPLC is designed with small particle stationary phases and shorted columns for achieving the recommended fastness during regular analysis in QC units. UHPLC offers a perfect performance if compared to the traditional HPLC instrument[20].

  1. Dong, M.W. and Zhang, K., Ultra-high-pressure liquid chromatography (UHPLC) in method development. TrAC Trends in Analytical Chemistry, 2014, 63, pp.21-30.

We hope the fulfillment of all reviewers’ comments and it’s our pleasure to perform any additional recommendations.

Reviewer 2 Report

This manuscript describes two methodologies based on thin layer chromatography (TLC) and ultra-high performance liquid chromatography (UHPLC) for the determination of two drugs, specifically, flumethasone pivalate (FP) and clioquinol (CL). This work contributes little novelty with respect to the study carried out by Sayed et al. (2014), where these two analytes were also analyzed by the same chromatographic techniques. The authors mention that they analyze them in the presence of phenoxyethanol preservative, but the methods were not validated for this analyte, only for FP and CL.

I have some comments for the paper, which must be taken into account to improve the article before it will be suitable for acceptance for publication.

-          Lines 57-59: indicate the regulation that establishes the limit (1223/2009?).

-          Lines 60-61: indicate what a heptane-1-sulphonic lytical approach consists of.

-          Lines 73-75: the information is repeated.

-          The first time that an acronym is mentioned, it should be fully written. For example, ICH, ALS, TCC, VL, QC, LOQ, LOD, Rts, etc.

-          Line 79: reference 21 does not match, maybe it is reference 19?

-          Line 81: the titles of the sections recommended in the journal template must be followed.

-          The same name for a compound/solvent must be used throughout the document. For example, acetonitrile or cyanomethane, methanol or methyl alcohol.

-          Line 91: indicate that it is a liquid chromatograph.

-          Section 2.1: authors should indicate the elution mode (isocratic or gradient), the injection volume, the flow, the column temperature, and the detection mode (PDA?).

-          The distribution of the sections is a bit chaotic. I recommend merging sections 2.2, 2.4 and 2.5 under the title “reagents and chemicals”. Move the information from lines 122-125 and 133-135 to section 2.5 where the preparation of the standard solutions and those used in the calibration curve are described. Move the information from lines 179-187 to the introduction section.

-          Explain why in section 2.6 it is indicated that 10 µL from each working solution was applied, but then, in section 2.7, 20 µL for ear drops and 40 µL/10 µL for cream were injected. Are not the standards supposed to be applied under the same conditions as the samples?

-          Lines 154-160: indicate the injection volume.

-          Line 175: if the united extracts (0.5mg of FP and 75 mg of CL) were either diluted with methanol to 50 mL final volume, the final concentrations would not be 0.01 mg/mL for FP and 1.5 mg/mL for CL?

-          Section 3.1: indicate the percentages in the MI and the volumes in the MII used. What is called reasonable separation in line 193? Perhaps it is better to show these optimization results in the supplementary material.

-          The format of the tables and their titles do not follow those indicated in the journal template.

-          Table 1: The table indicates that the TLC method takes 10 minutes, but in the text (line 128, 20 minutes were indicated). Indicate how the parameters were calculated (accuracy, precision, specificity, robustness, LOQ, LOD) in the materials and methods section). Indicate the units for LOQ and LOD.

-          Line 230-232: indicate the units of these concentrations.

-          Line 236: explain how was the recovery test carried out. The authors mention that they also included the preservative phenoxyethanol, but the table does not show these results for this compound.

-          Improve the quality of Figure 2 because the numbers of the axes cannot be read, and indicate what the abbreviation of the RF axis means.

-          Figure 4: It remains to indicate which one corresponds to a, b, c.

-          Tables 2 and 3: the standard addition technique was not explained in the materials and methods section. Did the authors use the samples that already contained the analytes to overload or did they perform this assay on a clean sample? What is the meaning of “claimed taken”? How the authors calculated the difference between total found and standard found? Why did the authors select those concentrations to add? The explanation of the superscript "a" at the bottom of Table 3 is missing to indicate that the 30 mg is for the CL.

-          Line 262 and line 270: indicate what are conventional and international standards, respectively.

-          Lines 266-268: where are these results?

-          Tables 4 and 6: explain how the parameters T, K', Rs and assay percentage were calculated.

-          The changes studied, for example, in the case of temperature or pH in the UHPLC method, are very slight, which is why there are hardly any variations.

-          Tables 5 and 7: explain how these parameters were calculated in the materials and methods section.

-          References 25-31: these articles are not of interest to the present work since the same analytes are not analyzed.

-          It remains to discuss the results obtained in the validation of the method (LOD, LOQ, etc.) with references 7-12 and 16.

-          Conclusions: the authors write about the new and innovative chromatographic methods developed but in reality, these same methods were already proposed in the previous study of 2014 (reference 16).

-          References: the references do not follow those indicated in the journal template.

Author Response

Replies to reviewers' comments

Great appreciation for all the reviewers and the editor for their productive recommendations. Really, the quality of this paper was greatly improved following their suggestions.   

Reviewer 2

This manuscript describes two methodologies based on thin layer chromatography (TLC) and ultra-high performance liquid chromatography (UHPLC) for the determination of two drugs, specifically, flumethasone pivalate (FP) and clioquinol (CL). This work contributes little novelty with respect to the study carried out by Sayed et al. (2014), where these two analytes were also analyzed by the same chromatographic techniques. The authors mention that they analyze them in the presence of phenoxyethanol preservative, but the methods were not validated for this analyte, only for FP and CL.

Thanks for your point of view. Our main aim was to analyze both drugs in the presence of phenoxyethanol preservative for the first time particularly in the cream dosage form. Therefore, we focus on the simultaneous assay and validation of both drugs only after their separation from the preservative that may interfere with their quantitative determination. In addition, the new UHPLC method has priority over the old HPLC one by Sayed et al., 2014 in terms of quickness and avoiding interference of the PEP preservative. This paper represents the first analytical approach for the concurrent assay of FP and CL in the presence of the preservative phenoxyethanol in the cream formulation.

I have some comments for the paper, which must be taken into account to improve the article before it will be suitable for acceptance for publication.

  1. Lines 57-59: indicate the regulation that establishes the limit (1223/2009?).

The following statement was added as suggested.

Phenoxy ethanol preservative (CAS n. 122-99-6) is reported in Annex V/29 of the Cosmetics Regulations for the European Commission, Scientific Committee on Consumer Safety, Phenoxy ethanol n. 1223/2009 [6].

  1. Lines 60-61: indicate what a heptane-1-sulphonic analytical approach consists of.

Sorry for this word error, corrected as recommended to be

many analytical approaches have been established for the assay of both drugs either individually or in combination with other drugs.

  1. Lines 73-75: the information is repeated.

Deleted as recommended.

  1. The first time that an acronym is mentioned, it should be fully written. For example, ICH, ALS, TCC, VL, QC, LOQ, LOD, Rts, etc.

Provided as suggested

International Conference for Harmonization (ICH) ; Automated Liquid Sampler (ALS),; Thermostated Column Copartment (TCC); Vertical In-Line Close (VL); Quality control (QC); Limit of detection (LOD) ; Limit of quantitation (LOQ) ; Standard Deviation (SD); Relative Standard Deviation (RSD); Retention Time (Rt); Retention times (Rts); internal diameter (I.D.)

  1. Line 79: reference 21 does not match, maybe it is reference 19?

Corrected thanks

  1. Line 81: the titles of the sections recommended in the journal template must be followed.

Corrected thanks

  1. The same name for a compound/solvent must be used throughout the document. For example, acetonitrile or cyanomethane, methanol or methyl alcohol.

Corrected as suggested

  1. Line 91: indicate that it is a liquid chromatograph.

Provided as suggested

  1. Section 2.1: authors should indicate the elution mode (isocratic or gradient), the injection volume, the flow, the column temperature, and the detection mode (PDA?).

The missed information was included

Separation was done on reversed phase inertsil ODS 5 µm C 18 stationary phase (4.6 × 50 mm, 100 Å) at room temperature, and isocratic elution was attained by acidic buffer pH 3 of phosphate type (having 100 mg Heptane-1-sulphonic acid sodium salt per 100 mL) and acetonitrile (35:65, by volumes). The injected volume was one microliter. The detection mode was a photodiode array detector (PDA).

  • The distribution of the sections is a bit chaotic. I recommend merging sections 2.2, 2.4 and 2.5 under the title “reagents and chemicals”. Move the information from lines 122-125 and 133-135 to section 2.5 where the preparation of the standard solutions and those used in the calibration curve are described. Move the information from lines 179-187 to the introduction section.

Done as suggested and references were renumbered.

  • Explain why in section 2.6 it is indicated that 10 µL from each working solution was applied, but then, in section 2.7, 20 µL for ear drops and 40 µL/10 µL for cream were injected. Are not the standards supposed to be applied under the same conditions as the samples?
  1. 10 µL from each working solution

 Amounts corresponding to 2.00-12.00 mg of FP and 2.00-10.00 mg of CL were moved from their corresponding parent flasks into two distinct series of 10-mL glass flasks

Upon transferring 1 mL from the stock solution 2 mg/mL into 10 mL flask, the concentration will be 0.2 mg/mL. Spot 10 µL gave 2 µg/band and upon transferring 6 mL from that stock into 10 mL flask the concentration will be 1.2 mg/mL so spot 10 µL gave 12 µg/band.

  1. 20 µL for ear drops:

Five milliliters of viotic® ear drop(one mL labeled to have 0.20 mg FP and 10.00 mg CL) was taken into a 10-mL  flask (0.10 mg per mL of FP and 5.00 mg per mL of CL ). 20.00 µL of this solution were spotted to determine FP (20 *0.1 mg =2 µg/band)

While CL was determined by diluting 0.4 milliliters of this solution into a 10-mL flask to give 0.2 mg/mL followed by spotting of 20.00 µL of this diluted solution to give 4 µg/band.

  1. 40 µL for cream:

2.5 g of locacorten vioform® cream (each one gram branded to have 0.2 mg of FP and 30 mg of CL) into 10-mL glass flask. To give (0.05 mg/mL of FP and 7.50 mg/mL CL) (MI). Aliquots of 40.00 µL of this solution were spotted to give 2.00 µg/band FP, then CL was determined by diluting 0.4 µL of this solution with methanol into 10-mL volumetric flask to give (0.3 mg/mL CL) followed by spotting 10.00 µL to give 3.00 µg/band CL.

  • Lines 154-160: indicate the injection volume.

Indicated as suggested

  • Line 175: if the united extracts (0.5mg of FP and 75 mg of CL) were either diluted with methanol to 50 mL final volume, the final concentrations would not be 0.01 mg/mL for FP and 1.5 mg/mL for CL?

Sorry for this unclear statement. 5 gram of the cream was used for extraction in MII ( UHPLC method while for M1, it was 2.5 gram. It was corrected in line 192 in the new version.

2.5 g of locacorten vioform® cream (each one gram branded to have 0.2 mg of FP and 30 mg of CL) into 10-mL glass flask. To give (0.05 mg/mL of FP and 7.50 mg/mL CL) (MI). Aliquots of 40.00 µL of this solution were spotted to give 2.00 ug/band FP, then CL was determined by diluting 0.4 µL of this solution with methanol into 10-mL volumetric flask to give (0.3 mg/mL CL) followed by spotting 10.00 µL to give 3.00 µg/band CL.

  • Section 3.1: indicate the percentages in the MI and the volumes in the MII used. What is called reasonable separation in line 193? Perhaps it is better to show these optimization results in the supplementary material.

The detailed percentages for the tested mobile systems were listed in Supplementary Table S1. Reasonable separation was replaced by optimal resolution that refers to optimal peak symmetry and highest achieved resolution

  • The format of the tables and their titles do not follow those indicated in the journal template.

The format of tables was followed as in the journal template as much as we can. However, we keep some of longitudinal line to prevent data interferences.

  • Table 1: The table indicates that the TLC method takes 10 minutes, but in the text (line 128, 20 minutes were indicated). Indicate how the parameters were calculated (accuracy, precision, specificity, robustness, LOQ, LOD) in the materials and methods section). Indicate the units for LOQ and LOD.

In table 1; it was mentioned that the actual analysis speed is 10 minutes for TLC method and 3 minutes for UHPLC method. However, in the text line 147 in the new version it was mentioned that “The separation tank was saturated for 20 minutes with the developing system containing benzene: ethyl acetate: formic acid (5: 5: 0.2, by volume) at 25 °C temperature.” Moreover, in table 4, it was mentioned that the saturation time (before actual analysis) was 20 minutes. In conclusion, the saturation time for mobile phase was 20 minutes while the actual analysis time was 3 minutes. The last statement was added for more illustration. Thanks

The detailed explanations about calculations of validation parameters were stated under section 3.2 Methods validation and they were highlighted. We think that repeating the same information in the methods section will be boring for readers.

The units for LOQ and LOD were provided in table 1.

  • Line 230-232: indicate the units of these concentrations.

Provided as suggested

  • Line 236: explain how was the recovery test carried out. The authors mention that they also included the preservative phenoxyethanol, but the table does not show these results for this compound.

The mean percent recovery (%R), for triplicate determinations of three concentration levels of each drug was calculated.

The word including was replaced by “ in the presence of “ as the preservative PE was not analyzed as the main aim for this work was to determine both drugs only.  Lines 248-250 in the new version.

  • Improve the quality of Figure 2 because the numbers of the axes cannot be read, and indicate what the abbreviation of the RF axis means.

Done as suggested. Thanks

  • Figure 4: It remains to indicate which one corresponds to a, b, c.

The symbols a, b, and C were added for Fig 4.

  • Tables 2 and 3: the standard addition technique was not explained in the materials and methods section. Did the authors use the samples that already contained the analytes to overload or did they perform this assay on a clean sample? What is the meaning of “claimed taken”? How the authors calculated the difference between total found and standard found? Why did the authors select those concentrations to add?

Standard addition technique is done by adding amount of standard drug to the dosage form. First we determine the amount of dosage form alone which is called claimed taken to check if it is matched the labeled amount in dosage form with a good recovery.

Second we add amount of standard drug to the claimed taken amount from dosage form and get the peak area then by substation in regression equation total concentration can be calculated so, by difference (Total concentration – claimed concentration)we can found the amount added from standard drug and the recovery can be calculated.

The explanation of the superscript "a" at the bottom of Table 3 is missing to indicate that the 30 mg is for the CL.

Corrected as suggested

  • Line 262 and line 270: indicate what are conventional and international standards, respectively.

Sorry for this typo error, the word conventional was replaced by the word “convenient “. Besides, reference number [25] was added for the international standards.

  • Lines 266-268: where are these results?

Illustrated as suggested

The recorded values for Retention times (Rts) of the medicines that illustrated in (Fig 3a) were relatively the same in all cases (Table 6) except for the flow speed, where slight changes in Rts were recorded. But, the calculated resolutions (Rs) were each time not less than 1.5, illustrating accepted chromatograms. Outcomes for Capacity (K') and tailing (T) factors were in alignment with international standards [23] (Table 6).

  • Tables 4 and 6: explain how the parameters T, K', Rs and assay percentage were calculated.

The explanations for calculations for the mentioned parameters were illustrated at the bottom of Table 4 and 6 as footer.

  • The changes studied, for example, in the case of temperature or pH in the UHPLC method, are very slight, which is why there are hardly any variations.

We try to follow ICH guidelines recommendation for robustness test as per ICH guideline robustness is defined as “The robustness of an analytical procedure is a measure of its capacity to remain unaffected by small, but deliberate variations in method parameters and provides an indication of its reliability during normal usage.”

  • Tables 5 and 7: explain how these parameters were calculated in the materials and methods section.

The explanations for calculations for the mentioned parameters were illustrated at the bottom of Table 5 and 7 as footer.

  • References 25-31: these articles are not of interest to the present work since the same analytes are not analyzed.

The old refs 25-31 was deleted and two relevant review articles were cited in the text and added for the reference list. Thanks

25.Ravi Sankar, P., Madhuri, B., Naga Lakshmi, A., Pooja, A., Bhargava Sai, M., Suresh, K. and Srinivasa Babu, P., 2020. Selected HPLC Applications-Quick Separation Guide: A Review. Int. J. Pharm. Sci. Rev. Res60(2), pp.13-20.

26.Ciura, K., Dziomba, S., Nowakowska, J. and Markuszewski, M.J., 2017. Thin layer chromatography in drug discovery process. Journal of Chromatography A1520, pp.9-22.

  • It remains to discuss the results obtained in the validation of the method (LOD, LOQ, etc.) with references 7-12 and 16.

Anew Table S2 supplementary material was included for comparisons of LOD and LOQ for the mentioned references. However, ref 7 was not included as Ref 7 was LC-MS/MS method for assay of flumethasone (not flumethasone pivalate) in calf urine and serum.

The following paragraph was added for the text lines 302-306 in the new version.

Upon comparisons of LOD and LOQ for our novel methods with old recorded methods in the literature (Table S2), our novel methods showed relatively comparable values with the methods stated by Sayed et al [16]. However, the maximal sensitivity was observed in the HPLC with electrochemical detection [12] by Bondiolotti. et. al., 2006 where it detected and quantified CL in the nano-gram level in plasma. 

  • Conclusions: the authors write about the new and innovative chromatographic methods developed but in reality, these same methods were already proposed in the previous study of 2014 (reference 16).

Conclusion was rewritten as recommended highlighting the merits for the new method over the previous study of 2014 (reference 16).

  • References: the references do not follow those indicated in the journal template.

All the references were checked and reformatted according to journal style.

Round 2

Reviewer 2 Report

The manuscript was modified according to the comments